# Anticandidal Activity of Capsaicin and Its Effect on Ergosterol Biosynthesis and Membrane Integrity of *Candida albicans*

**DOI:** 10.3390/ijms24021046

**Published:** 2023-01-05

**Authors:** Jawad M. Behbehani, Mohammad Irshad, Sheikh Shreaz, Maribasappa Karched

**Affiliations:** 1Department of Restorative Sciences, Faculty of Dentistry, Kuwait University, Kuwait City 13060, Kuwait; 2Department of Bioclinical Sciences, Faculty of Dentistry, Kuwait University, Kuwait City 13060, Kuwait; 3Dasman Diabetes Institute, Dasman 15462, Kuwait

**Keywords:** chili pepper, Capsaicin, anticandidal, ergosterol

## Abstract

Oral candidiasis is an infection of the oral cavity commonly caused by *Candida albicans*. Endodontic treatment failure has also been found to be persistent from *C. albicans* in the root canal system. Despite the availability of antifungal drugs, the management of *Candida* oral infection is difficult as it exhibits resistance to a different class of antifungal drugs. Therefore, it is necessary to discover new antifungal compounds to cure fungal infections. This study aimed to examine the antifungal susceptibility of Capsaicin, an active compound of chili pepper. The susceptibility of Capsaicin and Fluconazole was tested against the *Candida* species by the CLSI (M27-A3) method. The effect of Capsaicin on the fungal cell wall was examined by the ergosterol inhibitory assay and observed by the scanning electron micrograph. The MIC range of Capsaicin against *Candida* isolates from oral (n = 30), endodontic (n = 8), and ATCC strains (n = 2) was 12.5–50 µg/mL. The MIC range of Fluconazole (128- 4 µg/mL) significantly decreased (2- to 4-fold) after the combination with Capsaicin (MIC/4) (*p* < 0.05). Capsaicin (at MIC) significantly reduced the mature biofilm of *C. albicans* by 70 to 89% (*p* < 0.01). The ergosterol content of the cell wall decreased significantly with the increase in the Capsaicin dose (*p* < 0.01). Capsaicin showed high sensitivity against the hyphae formation and demonstrated a more than 71% reduction in mature biofilm. A fluorescence microscopy revealed the membrane disruption of Capsaicin-treated *C. albicans* cells, whereas a micrograph of electron microscopy showed the distorted cells’ shape, ruptured cell walls, and shrinkage of cells after the release of intracellular content. The results conclude that Capsaicin had a potential antifungal activity that inhibits the ergosterol biosynthesis in the cell wall, and therefore, the cells’ structure and integrity were disrupted. More importantly, Capsaicin synergistically enhanced the Fluconazole antifungal activity, and the synergistic effect might be helpful in the prevention of Fluconazole resistance development and reduced drug-dosing.

## 1. Introduction

*Candida albicans* is a commensal fungus, and about 50% of the global population has it in their oral cavity without exhibiting infection [1]. However, overgrowth causes infection of the oral cavity, commonly known as candidiasis [2]. It also causes severe infections in immunocompromised hosts or those whose natural flora has been altered [3]. The clinical sample of the Kuwait Dental Clinic also contained many *Candida* species; however, *C. albicans* was prominent [4]. *C. albicans* is a filamentous fungus that easily forms a colony in the oral cavity on the tongue, periodontal pockets, supragingival, periodontal pocket mucosa, and dentin and invades the underlying tissue [3,5]. *C. albicans* also enters into the root canal system through dental carries and poses a significant challenge during the treatment of patients [6]. Most of the endodontic treatment failure has been reported due to the persistence of *C. albicans* in the root canal system [7]. Many root canal irrigant solutions have been used in clinics to remove root canal infections. However, some fungi, such as *C. albicans*, resist the antibacterial agents present in the irrigant solution [7]. On the other hand, the frequent use of irrigant solutions raises serious clinical concerns because of their toxicity to periapical tissue, damage to permanent tooth follicles, and potential to produce acute allergic reactions [8]. 

Fluconazole has commonly been used as a principal and first-line drug in cases of oral *Candida* infection. The clinician also recommends other antifungal drugs such as itraconazole or ketoconazole. Due to the indiscriminate use of these drugs in clinical therapy, resistance is emerging against most antifungal drugs [9]. Amphotericin-B is known as a gold standard antifungal drug, but it is limited to clinical use due to nephrotoxicity and other side effects in the recipient [10]. This situation underlies the need for safe, novel, and effective antifungal compounds. 

A wide range of natural molecules has been identified as the most successful source of therapeutic drugs [11,12]. Many studies reported the antifungal and antimicrobial properties of chili pepper extract [13,14]. Similarly, the extract of Capsicum Chinese was reported to have antifungal and antiparasitic activities [15,16]. Capsaicin (N-vanillyl-8-methyl-6-(E)-noneamide) is one of the primary active compounds of chili peppers (genus capsicum) [17]. Since ancient times, the chili pepper has been used as a spice in food worldwide. In green and red peppers, the content of Capsaicin ranges from 0.1% to 1% [18]. In addition, the oral toxicity of Capsaicin was reported in mice and rats (about 97 mg/kg) in a high concentration [19]. In this study, we examined the antifungal activity of Capsaicin against 30 oral and 8 endodontic *Candida* isolates. We also used various techniques to study the fungicidal effects of Capsaicin. Finally, we studied the synergistic effects of Capsaicin along with the Fluconazole drug on the susceptibility of *C. albicans*. A combination effect of two antifungal agents might be a possible strategy in the future to combat the growth of the *Candida* species. 

## 2. Results

### 2.1. Minimum Inhibitory Concentration (MIC)

Capsaicin showed antifungal activities against clinical *Candida* isolates and the ATCC strains. The MIC ranges of Capsaicin against the 30 oral isolates were 12.5 to 50 µg/mL, against the 2 ATCC strains were 12.5 to 25 µg/mL, and against the 8 endodontic isolates were 25 µg/mL. The MIC ranges of Fluconazole and Ketoconazole were 16 to 128 µg/mL and 8 to 16 µg/mL, respectively (Table 1). However, the MIC of Capsaicin and Fluconazole against the ATCC strain (*C. albicans*) was 25 and 64 µg/mL, respectively. The MIC ranges of Capsaicin and Fluconazole against the different Candida species isolated from the oral sample are given in Appendix A.

### 2.2. Synergistic Activity of Capsaicin with Fluconazole

Various combinations of Capsaicin and Fluconazole were used to test the combined effect against the endodontic isolates. The combination of Capsaicin with Fluconazole showed a remarkable decrease in the MIC values compared to each drug alone (Table 2). The MIC of Fluconazole significantly decreased from 16–128 to 4–16 µg/mL, whereas the MIC of Capsaicin significantly decreased from 25 to 6.25 or 12.5 µg/mL (*p* < 0.01). The combinations of Capsaicin and Fluconazole exhibited synergism effects (FICI ≤ 0.50). The combination did not show an antagonistic effect against the tested isolates. The antiCandidal activities of Capsaicin and Fluconazole alone and in combination against the 30 oral isolates are given in Appendix A. 

### 2.3. Biofilm Inhibition

The effect of Capsaicin (MIC) was tested on the pre-formed mature biofilm of *C. albicans*. In Appendix A, the left panel represents the CLSM image of the mature biofilm of *C. albicans*, whereas the right panel represents the biofilm morphology after the treatment. The CLSM image clearly showed the eradication of biofilm of the treated sample. The metabolic activity of biofilm assessed by the MTT (3-(4,5-dimethylthiazol-2-yl)-2,5-diphenyltetrazolium bromide) metabolic reduction assay demonstrated an average 73.6% reduction in the biofilm of endodontic *Candida* isolates, while for ATCC strains, it was 71.2% at the MIC level (Figure 1). In summary, our data indicate that Capsaicin significantly inhibits the mature biofilm (*p* < 0.01).

### 2.4. Inhibitory Activity on Hyphae Formation 

A hyphae-inducing medium was used to examine the hyphae inhibition by Capsaicin. The live cell observer microscope recorded live images of cells’ transitions from yeast form to filaments form. In Figure 2, the left panel shows the untreated *C. albicans* (control) cells, and the right panel shows the cells’ morphology after the treatment with Capsaicin (MIC). The untreated controls showed massive hyphae growth after three hours, but in the presence of Capsaicin (MIC), hyphae growth was not observed. Cells treated with Capsaicin showed minor changes in cell morphology. The results clearly showed that Capsaicin inhibits the growth of hyphae.

### 2.5. Inhibition Activity of Ergosterol Synthesis

The ergosterol content of treated and untreated *C. albicans* cells’ membranes was estimated. In Figure 3, the flat curve represents the significant reduction in ergosterol content in Capsaicin (MIC)- and Ketoconazole (64 µg/mL)-treated cells (*p* < 0.05). However, untreated control cells represented the sharp peaks of ergosterol. The mean reduction in the ergosterol content of the cells treated with MIC/4, MIC/2, and MIC of Capsaicin and 64 µg/mL of Ketoconazole was 42.4%, 88.7%, 98.6%, and 99.1%, respectively. The dose-dependent decrease in ergosterol content indicated that the antifungal effect of Capsaicin was dependent on its concentrations, and its primary mode of action was through the inhibition of ergosterol biosynthesis. 

### 2.6. Cell Membrane Permeability

A confocal laser scanning microscope (CLSM) was used to investigate individual *C. albicans* cells stained with propidium iodide. The CLSM result showed the membrane-impermeant propidium iodide uptake by the Capsaicin-treated *Candida* cells. Propidium iodide is a red color intercalating stain that is membrane-impermeant and is therefore excluded by healthy cells. In Figure 4, CLSM images in the left panel show the untreated *C. albicans* that did not take up the red dye, whereas the dye was taken up by the treated cells in the right panel. Our results confirm that propidium iodide penetrates Capsaicin-treated cells, suggesting that Capsaicin disrupted the structure of the cell membrane.

### 2.7. Change of Cellular Morphology

The electron micrographs of Capsaicin-treated cells obtained from the SEM showed critical morphological changes. The monograph of *C. albicans* cells was treated with MIC values of Capsaicin (Figure 5). In the monograph of the SEM, untreated *Candida* cells appeared to be oval and with smooth cell surfaces and polar bud scars (Figure 5A–C). While cells treated with Capsaicin at its respective MIC values for 12 h resulted in deformed cells with convoluted and irregular surfaces, a deposit of lytic material in the form of vesicles was also noticed. The treated cells showed a complete change in cell shape due to the formation of deep furrows and wrinkles on the cell surface (Figure 5D–F).

## 3. Discussion

The mechanism of action of antifungal compounds has recently been given much attention by researchers. Furthermore, it is expected that the combination of the natural antifungal compounds with the antimycotics might provide promising rapid and synergistic efficacy and help to pull down the risk of drug resistance and antimycotics’ drug-dosing toxicity. In the present study, Capsaicin provided potential antifungal activity, as tested against *Candida* isolates and ATCC reference strains. Capsaicin inhibited the growth of *C. albicans,* as observed by the different assays. However, the MICs against *C. albicans* varied among the isolates. The MIC value of Capsaicin against *Candida* species is low compared to the previous reports tested against *Streptococcus pyogenes* [20], and *Porphyromonas gingivalis* [21]. To the best of our knowledge, this is the first study that reported the antifungal activity of the Capsaicin compound. 

Our study represents the first evidence of the combined effects of Capsaicin and Fluconazole antifungal drugs against *C. albicans.* The treatment of the Fluconazole drug along with Capsaicin showed a decrease in the MIC value of Fluconazole in comparison to the effect of the drug alone (Table 2). The eradication of the *Candida* biofilm has become one of the most challenging tasks in fungal therapeutics because of its reduced susceptibility to antimycotic drugs compared to planktonic cells [22]. In this study, Capsaicin showed a strong ability to destroy the mature biofilm of *C. albicans* at the MIC concentration. This ability is also consistent with the inhibitory activity of Capsaicin on the hyphae formation (Figure 2). Hyphae development and biofilm formation are closely interlinked in *C. albicans.* Hyphae play a vital role in biofilm formation and across linked biofilm morphological integrity [23]. Secondly, the interface of Capsaicin with the hyphae formation can be attributed to the disturbance in the *Candida* cell membrane and metabolism, probably affecting fungal cell wall ergosterol biosynthesis enzymes. Ergosterol is an essential constituent of the fungal membrane for maintaining cell integrity, membrane fluidity, and cell metabolism and has been targeted for antifungal drug discovery [24]. In this study, the ergosterol content decreased in the treated cells with the increased Capsaicin concentration. The results referred to one of the primary modes of action of the Capsaicin compound. The decrease in ergosterol content in the cell membrane may create polar pores in fungal cell membranes and a consequent loss of essential ions such as potassium and protons and other molecules, ultimately killing the fungal cells [24,25]. A confocal laser microscopy showed the intensity of red propidium iodide (a fluorescent dye) inside the dead cells or cells with compromised membranes because it is not taken up by cells with an intact plasma membrane (Figure 4). This is the second line of evidence that the fungicidal activity of Capsaicin on endodontic yeasts might be due to a severe lesion on the membrane or loss of cell membrane integrity [26]. The third piece of evidence was revealed by the SEM micrographs. The untreated cells had a typical oval shape, smooth cell surface, and polar bud scars (Figure 5A–C). However, the treated cells showed a deformed cell shape and receding of the cytoplasm, leading to lysis of the cells (Figure 5D–E). Such surface alterations have been observed earlier in the case of *C. albicans* treated by antifungal agents [27,28]. The overall collapse of the cell wall is indicative of permeability changes that apparently provoke osmotic imbalance and ultimately result in cell death. 

## 4. Material and Methods

### 4.1. Chemicals

Capsaicin (purity ≥ 95%), Fluconazole, Ketoconazole, chemicals, and culture medium constituents were purchased from Sigma-Aldrich, St. Louis, MO, USA.

### 4.2. Sample Collection

Oral *Candida* species samples were collected from dental patients at the Kuwait University Dental Clinic (KUDC) using a standard oral rinse technique [29]. The collected samples were identified by using the CHROMagar medium (CHROMagar *Candida*, France) and VITEK-2 system assays (bioMerieux, Craponne, France), as listed in Appendix A. Endodontic *C. albicans* isolates (n = 8) were obtained from the Oral Microbiology Laboratory, Faculty of Dentistry, Kuwait University, Kuwait. 

### 4.3. Minimum Inhibitory Concentration 

The Clinical Laboratory Standards Institute (CLSI) for yeasts (M27-A3) method was used to estimate the minimum inhibitory concentration (MIC) of the test compound and the drug [30]. The MIC was determined in an RPMI-164 medium (Cat no. R6504, Sigma-Aldrich, St. Louis, MO, USA), and buffered to pH 7.0 with 0.165 M morpholinepropanesulfonic acid (MOPS). The overnight culture of *Candida* cells was diluted in media, and a 100 μL volume of this diluted inoculum was added to each well of the 96-well U-bottom tissue culture plate, resulting in a final inoculum of ~2.5 × 10^3^ cells/mL. The concentration range tested was 1.56 to 400 µg/mL for Capsaicin and 0.50–128 µg/mL for Fluconazole and Ketoconazole. The medium without the drugs was used as the control, and the blank control used contained only the medium. The plates were incubated at 37 °C for 48 h in the incubator. The plates were read visually, and the MIC was defined as the lowest concentration of the test agents that prevented visible growth with respect to the growth control. All the assays were performed in triplicate.

### 4.4. Synergistic Activity of Capsaicin and Fluconazole 

Synergistic activity was measured by the checkerboard microtiter plate method [30,31]. Capsaicin and Fluconazole drugs were serially diluted in RPMI at different combinations. The final concentrations ranged from 1.56 to 100 µg/mL for Capsaicin and from 1 to 64µg/mL for Fluconazole. *Candida* cells were added to each well at a final concentration of ~2.5 × 10^3^ cells/mL and incubated for 48 h at 37 °C in the incubator. The growth of *Candida* cells was read visually. All assays were performed in triplicate. 

The fractional inhibitory concentration index (FICI) was used to evaluate the interactions between Capsaicin and drugs. FICI = MIC (Capsaicin with Fluconazole)/MIC (Capsaicin alone) + MIC (Fluconazole with Capsaicin)/MIC (Fluconazole alone). FICI ≤ 0.5 indicated synergetic interaction between Capsaicin and Fluconazole, FICI > 0.5 to 4 indicated no interaction, and FICI > 4 indicated antagonistic interaction between both agents.

### 4.5. Biofilm Inhibition Assay

A standardized protocol was used in this study [4]. The mature biofilms of *Candida* were formed in the wells of polystyrene 96-well microtiter plates (Corning Inc., Corning, NY, USA). The MIC of Capsaicin was prepared separately in an RPMI 1640 medium and added to each well (200 µL/well) and incubated for 24 h at 37 °C. After the incubation period, the medium was aspirated and washed carefully with sterile PBS (pH 7). The density of biofilm was observed under confocal laser microscopy (Zeiss LSM 500) by using Cyto-9 fluorescence dye. The biofilm density was quantified by the MTT assay, and the percentage of biofilm inhibition was calculated [25]. All experiments were performed in triplicate.

### 4.6. Study of Hyphae Formation and Cells Morphology 

The aliquot of overnight culture cells (~10^6^ cells/mL) inoculated in YPD medium was supplemented with 10% fetal bovine serum (FBS) [23]. The MIC of Capsaicin was added into the medium and kept at 37 °C under the Eye lens (20 x) of the cell observer microscope (Zeiss LSM). The morphological change of *Candida* cells was live-recorded every 15 min. Cells without treatment were used as a positive control. 

### 4.7. Ergosterol Extraction and Estimation 

The total intracellular sterols were extracted by a standard method [32], with arrangements as described in our previous work [33]. *Candida* colony from an overnight Sabouraud dextrose agar plate culture was used to inoculate 10 mL of Sabouraud dextrose broth for the control and various concentrations of Capsaicin. Ketoconazole was used as a positive control. The cultures were incubated for 16 h and harvested by centrifugation at 4000× *g* for 5 min. The net weight of the cell pellet was determined. In all, 3 mL of 25% ethanolic potassium hydroxide solution was added to each pellet and vortex-mixed for 3 min. Cell suspensions were transferred to sterile borosilicate glass screw-cap tubes and were incubated at 85 °C in a water bath for one hour and then allowed to cool at room temperature. Sterols were then extracted by the addition of a mixture of 1 mL of sterile Milli-Q water and 3 mL of n-heptane followed by vigorous vortex mixing for 3 min. The *n*-heptane layer was transferred to a clean borosilicate glass screw-cap tube and stored at −20 °C. 

The concentration of sterol was scanned spectrophotometrically between 230 and 300 nm with a spectrophotometer (Eppendorf, USA). The ergosterol content was calculated as a percentage of the wet weight of the cell using formula
% Ergosterol + %24 (28) dehydroergosterol (DHE) = [(A281.5/290)/pellet weight]
%24(28) DHE = [(A230/518)/pellet weight]
%ergosterol = [% ergosterol + %24(28) DHE] [%24(28) DHE]

The values 290 and 518 are the E values (in percentages per centimeter) determined for crystalline ergosterol and 24 (28) DHE, respectively.

### 4.8. Confocal Scanning Laser Microscopy (CSLM)

The disruptive effect of Capsaicin on *Candida* cell membranes was assessed using fluorescence dye Propidium iodide (PI) [4]. *Candida* cells (~1 × 10^6^) were suspended in a YPD medium and incubated with Capsaicin at 37 °C with constant shaking (120 rpm) for 12 h. To confirm cell membrane permeabilization, 1 µg/mL of dye was added, and the cell suspensions were then incubated again at 37 °C for 30 min. The dye uptake was visualized by confocal microscopy (Zeiss LSM 500). 

### 4.9. Effect of Capsaicin on the Biofilm of Dentine Substrate 

Freshly extracted teeth were collected from the Kuwait Dental Clinic. The teeth pulp chamber was opened by using a cutting machine (Accutom-100, USA). The teeth slice was washed and sterilized by the treatment with 17% EDTA, followed by 2.5% sodium hypochlorite for 4 min, and then air-dried for 10 min. The teeth slice was placed into the freshly prepared *C. albicans* cells’ suspension (1 × 10^6^) in YEPD medium and incubated with shaking (120 rpm) at 37 °C for 24 h. After the incubation period, the teeth slice was transferred to the fresh medium that contained the test molecule (MIC) and re-incubated for 24 h. After the end of the treatment period, the dental slice was washed thrice carefully with PBS and processed for scanning electron microscopy (JEOL, Carryscope JCM 5700).

### 4.10. Statistical Analysis

The experiments were performed in triplicate. A statistical analysis was performed by SPSS software (SPSS Inc., USA). The significant differences between the groups were analyzed by *t*-test. The value *p* ≤ 0.05 was considered a statistically significant difference.

## 5. Conclusions

Capsaicin shows high antifungal activity against *C. albicans*. Capsaicin inhibits the hyphae formation, and the ergosterol biosynthesis pathway provides strong evidence for the antifungal potential. Capsaicin plays a crucial role in biofilm inhibition. In addition, the synergistic effect of Capsaicin with Fluconazole might be helpful in reducing the dosage of antimycotics with the benefit of avoiding drug resistance and side effects. This study provides various evidence of the fungicidal potential of Capsaicin molecules. Based on the result, we believe that Capsaicin could be a promising molecule against oral fungal infection.

## Figures and Tables

**Figure 1 ijms-24-01046-f001:**
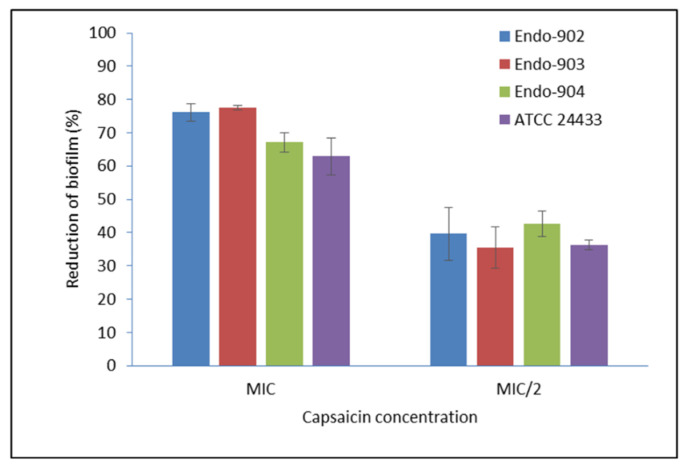
Effect of Capsaicin on the mature biofilm of *C. albicans.* The standard deviations (SD) of each sample are shown in the graph. The mean differences between the control and test molecules at the MIC level were statistically significant (*p*-level, 0.01). MIC/2—half of the MIC concentration; Endo—endodontic isolates.

**Figure 2 ijms-24-01046-f002:**
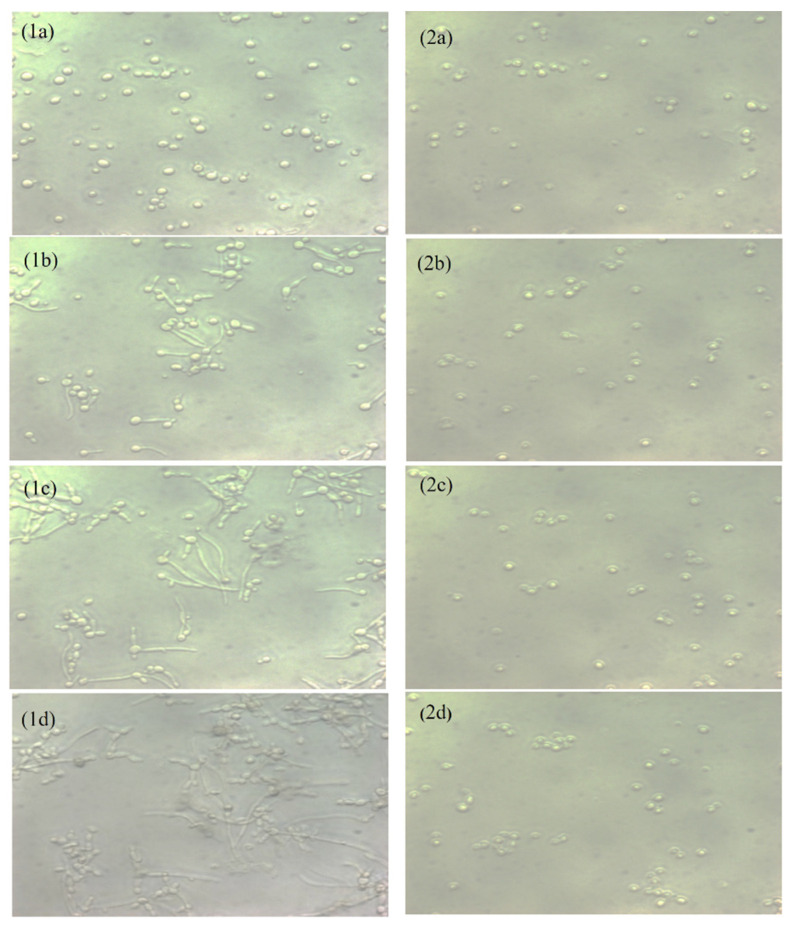
Effect of Capsaicin (MIC) on the hyphae growth of *C. albicans* (Endo-902). The live cells photograph was recorded by the cell observer microscope (20×). The photograph of untreated cells (**1a**–**1d**) and treated cells (**2a**–**2d**) were recorded at three-hour time points, respectively.

**Figure 3 ijms-24-01046-f003:**
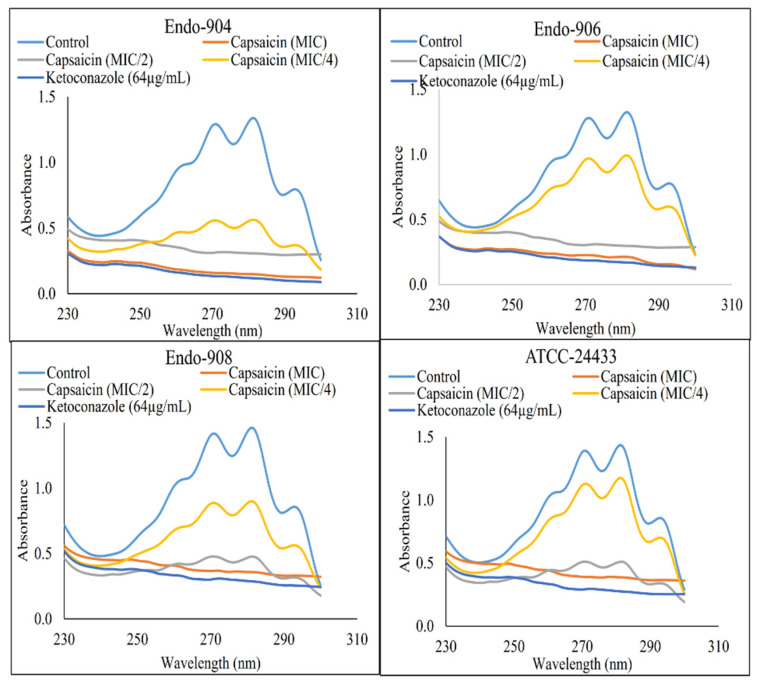
Spectrophotometric sterol profiles of *C. albicans* (Endo-904, Endo-906, and Endo-908). Sterols were extracted from the cells and spectral profiles between 230 and 300 nm were determined.

**Figure 4 ijms-24-01046-f004:**
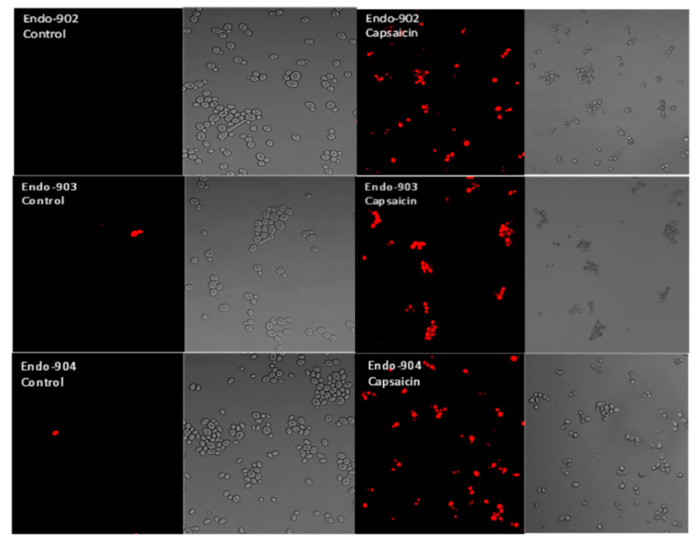
Confocal scanning laser microscopy (CSLM) images of endodontic *C. albicans* cells (Endo) treated with Capsaicin (MIC) (right panel) and untreated control cells (left panel). To observe membrane damage, cells were stained with propidium iodide (red signals).

**Figure 5 ijms-24-01046-f005:**
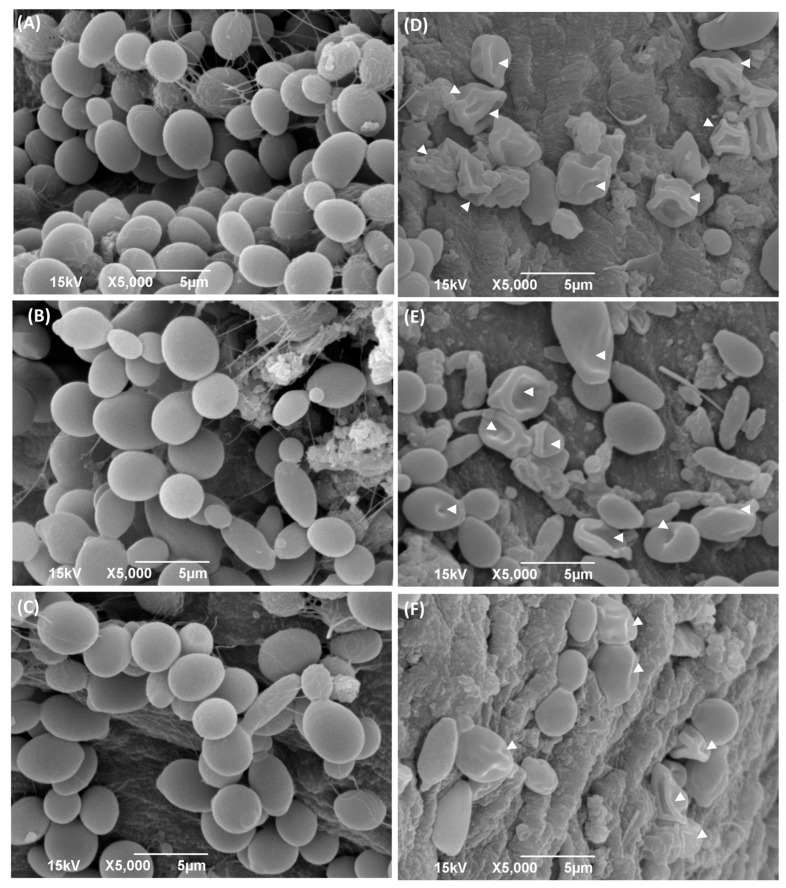
Scanning electron micrograph of *C. albicans* cells with and without treatment with Capsaicin (MIC). Attachments of *Candida* cells (Endo-902 (**A**), Endo-903 (**B**), and Endo-904 (**C**)) with teeth pulp in without treatment condition are shown in left panel, whereas with treatment condition (Endo-902 (**D**), Endo-903 (**E**) and Endo-904 (**F**)) are shown in right panel.

**Table 1 ijms-24-01046-t001:** MIC values of Capsaicin against *C. albicans*.

*C. albicans* (Strain/Isolates)	MIC (µg/mL)		
	Capsaicin	Fluconazole	Ketoconazole
*ATCC-24433*	25	64	16
Isolates			
Endo-902	25	64	16
Endo-903	25	64	16
Endo-904	25	128	8
Endo-905	25	64	16
Endo-906	25	128	16
Endo-908	25	32	8
Endo-910	25	16	8
Endo-911	25	128	16
*Oral isolates#*	12.5–50	18–128	--

**ATCC—**American type culture collection; **Endo—**endodontic isolates; **MIC**—minimum inhibitory concentration.

**Table 2 ijms-24-01046-t002:** Antifungal effects of Capsaicin alone and in combination with Fluconazole against *C. albicans*.

*C. albicans* (ATCC/Isolates#)	MIC (µg/mL)				
	Capsaicin		Fluconazole		Σ FICI
	Alone/Combo	FICI	Alone/Combo	FICI	
*ATCC-24433*	25/6.25	0.25	64/4	0.063	0.313
Endo-902	25/6.25	0.25	64/16	0.250	0.500
Endo-903	25/6.25	0.25	64/8	0.125	0.375
Endo-904	25/12.5	0.50	128/16	0.125	0.625 *
Endo-905	25/6.25	0.25	64/4	0.063	0.313
Endo-906	25/6.25	0.25	128/16	0.125	0.375
Endo-908	25/6.25	0.25	32/8	0.250	0.500
Endo-910	25/12.5	0.50	16/8	0.500	1.000 *
Endo-911	25/12.5	0.50	128/16	0.125	0.625 *

**ATCC—**American type culture collection; **Endo—**endodontic isolates; **FIC**—fractional inhibitory concentration index; * no interaction; synergistic test result against the oral isolates given in Appendix A.

## Data Availability

Not applicable.

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
