# Peer review of "Anticandidal Activity of Capsaicin and Its Effect on Ergosterol Biosynthesis and Membrane Integrity of Candida albicans"

_ijms, 2023, doi:10.3390/ijms24021046_

Round 1

Reviewer 1 Report

This is a through and original study of antifungal effect of capsaicin on planctonic and sessile Candida cells.

Please check attached text for questions and suggestions for clarification of content.

General:

Please clarify number and species of the isolates included the study and how/why they were chosen. Title only states C.albicans but oral isolates mentioned  in the material/methods and Supplementary table includes other species. 

If only C.albicans were included, please prefer C.albicans instead of Candida throughout the text.

If species other than C.albicans were included, please group isolates according to species as fluconazole susceptibility varies. In my opinion, presentation of study data would improve when different species were evaluated seperately. Especially effect on species with intrinsic resistance/reduced susceptibility to to fluconazole (such as C.krusei and C.glabrata) would be interesting.

Please clarify if fluconazole resistant C.albicans isolates were chosen as almost all of them were resistant. Fluconazole resitance ratio among C.albicans is not expected to be that high.

Please check organism names, genus  and species name should be in italics, genus names should begin with a capital letter.

Should "Capsaicin" be capitalized?

Abstract:

You might consider underlining the effect on biofilms in the abstract (and title perhaps?)

Please add a brief description of the methods other than CLSI M27-A3 in the abstract before results. No mention of biofilm testing or ergosterol content. 

Materials and methods

Were all tests performed on all isolates? If not, please clarify test/microorganism combinations when presenting results. Any reasons to perform on a selected group of isolates?

Quality control strains C.krusei ATCC 6258 and C.parapsilosis ATCC 22019 are needed for antifungal susceptibility testing (CLSI M27-S4 as referenced in CLSI M27-A3 and the more current versions). No mention of any of these two quality control strains in the text. C.albicans ATCC 24433 is recommended for MACROdilution.

Detailed description of the methods, if they are well known (and described in detail in thre reference, such as CLSI M27-A3) may be avoided to help focusing on original content.

Candida biofilm inhibition assay is only performed for capsaicin. Fluconazole is not expected to have antibiofilm effect by itself. As capsaicin combination decreases MICs of planctonic cells for both , it might be interesting to see if synergistic effect is alos observed in biofilms.

Please add method for hyphae formation (germ tube test?)

Please consider order of methods in the text. Mentioning biofilm tests consecutively in material/methods and results sections after tests on plantonic cells might be less confusing.

Discussion:

Toxicity of capsaicin were not tested. I believe that it might be a major disadvantage for in vivo use of the molecule. Please include in discussion section.

Capsaicin in vitro antifungal activity was reported previously, especially for plant pathogens.  For example: DOI: 10.3390/antibiotics11091154, DOI: 10.1080/14786419.2018.1514395, DOI: 10.3390/polym14142774. This is a more thorough study.

All figures and tables should be understood clearly without referring to the text. Table 1 has MIC data of all endo isolates which are all C.albicans. Please clarify number and species of oral isolates.

Table 2 has data on endo isolates only. What about oral isolates?

Please provide a legend for supplementary figure.

Supplementary table: Data on endo isolates are also given in Table 2. Please avoid duplications or explain. Data on only 30 oral isolates were given, what about the others?

Author Response

We sincerely thank the reviewer for his/her valuable time on critically assessing our manuscript. We believe that the comments and suggestions have helped us improve the manuscript.

Reviewer 2 Report

The study aims to evaluate the anticandidal activity of capsaicin through single or in combination with fluconazole susceptibility tests for the management of oral infections.

The manuscript need deep remodulation expecially in the introduction and conclusions.

The evaluation for species other than Candida is poor and would need to be implemented.

Strenght: the anticandidal activity of capsaicin on C. albicans and its effect on ergosterol biosynthesis and membrane integrity are promising

Major revisions:

Results referred to the supplementary table 1 and statistical analysis  are missing.

Line 71: Results referred to species other than C. albicans are not reported.

C. albicans should be separated from different specie and results reported separately for a given specie.

Additional strains tested for species other than Candida would add value to the combination studies.

Why were the plates incubated for 24-48h? 

Introduction: Add information on Candida species responsible for oral infections other than Candida with data from literature and also from the authors local experience.

Explain why did the authors focused their attention on C. albicans.

Add information on antifungals commonly used with differences existing basing on the different Candida species or long therm therapies.

Line44: A wide range of natural molecules... please cite some examples with references. Add studies referred to natural molecules used as oral fungicidal.

minor revisions:

Supplementary figure 1: caption missing

Line 50: Esplicate acronymous when reported for the first time in the text

Line 83: Esplicate MTT reduction assay, please

Line 85: Elucidate the difference between MIC and MIC/2 capsaicin concentration.

Line 263: Esplicate the acronymous DHE

Line 148: Candida needs to be capitalized and italicized

Line 261: Add spectrophotometer name and basic setting parameters used

Sample collection: In which period of time were the samples collected and how were they conserved

Line 218: Add references with respect to the cut-off used

Materials and methods:

Change Candida with C. albicans where appropriate

Conclusions

Line 284: Only results referred to supplementary table 1 include specie other than Candida so please change the conclusions accordingly.

Add considerations about the novelty of the study and recap important results, expecially on biofilm disruption and its importance

Author Response

We sincerely thank the reviewer for his/her valuable time on critically assessing our manuscript. We believe that the comments and suggestions have helped us improve the manuscript. Kindly see the attachment for our responses to each comment.
